# High-Temperature Degradation Tests on Electric Double-Layer Capacitors: The Effect of Residual Voltage on Degradation

**DOI:** 10.3390/ma14061520

**Published:** 2021-03-20

**Authors:** Tomoki Omori, Masahiro Nakanishi, Daisuke Tashima

**Affiliations:** 1Department of Electrical Engineering, Graduate School of Engineering, Fukuoka Institute of Technology, Fukuoka 811-0295, Japan; mem18105@bene.fit.ac.jp; 2Department of Electrical Engineering, Faculty of Engineering, Fukuoka Institute of Technology, Wajiro-higashi 3-30-1, Higashi-ku, Fukuoka 811-0295, Japan; m-nakanishi@fit.ac.jp

**Keywords:** supercapacitor, electric double-layer capacitor, heat degradation, residual voltage

## Abstract

The demand for electric double-layer capacitors, which have high capacity and are maintenance-free, for use in a variety of devices has increased. Nevertheless, it is important to know the degradation behavior of these capacitors at high temperatures because they are expected to be used in severe environments. Therefore, degradation tests at 25 °C and 80 °C were carried out in the current study to analyze the degradation behavior. Steam-activated carbon, Ketjen black, and PTFE were used as the electrodes, conductive material, and binder, respectively, and KOH was used as the electrolyte. The impedance and capacitance were calculated from the voltage and current in the device using the alternating current (AC) impedance method. The results showed that the impedance increased and the capacitance decreased over 14 days at 80 °C, which is the inverse of what we observed at 25 °C. Rapid degradation was also confirmed from the 80 °C degradation test. The residual voltage after measuring the current and voltage was a prominent factor influencing this rapid degradation.

## 1. Introduction

In recent years, the demand for electric double-layer capacitors (EDLCs), also known as supercapacitors or ultracapacitors, has been increasing. EDLCs are characterized by their extremely low internal resistance and the fact that charging and discharging are not driven by chemical reactions. These characteristics enable rapid charge or discharge cycles and a long cycle life [1,2,3,4]. Owing to such advantages, EDLCs have been used in vehicles for regenerative energy recovery and as storage devices for renewable energy [5,6,7,8,9]. However, EDLCs possess low energy densities [10]. To address this drawback, research has been conducted to determine the most suitable materials and operating conditions to optimize the energy density of EDLCs. 

One possible solution is to increase the operating voltage [10,11]. However, in addition to high temperatures, using voltages higher than the rated voltage (1 V for aqueous EDLCs and 2.5 to 3 V for organic EDLCs) accelerates the degradation of EDLCs [12,13,14,15,16,17,18]. Various degradation tests involving high temperatures and voltages have been reported previously. One study showed that a 100% increase in impedance (*Z*) or a 20% decrease in capacitance (*C*) affects the performance of EDLCs [19]. 

To accurately observe this degradation criteria, it is important to understand the degradation behavior and observe the exact initial values of *Z* and *C*. Porous activated carbon is often used as the electrode material for EDLCs. The electrolyte solution cannot completely penetrate the pores immediately after the electrode is placed in it, and the measurement values of *Z* and *C* would be unstable. To obtain accurate initial values, measurements must be taken after complete penetration, and the time required for penetration must be known. Further, the voltage applied during the measurements should be considered. Accelerated degradation may still occur even if the applied voltage is left in a steady state. 

In this study, degradation tests were conducted at room temperature (25 °C) and high temperature (80 °C) (expected to be used in vehicles) [8], and the effect of residual voltage on the initial and terminal behaviors of *Z* and *C* was investigated. Cyclic voltammograms and galvanostatic charge/discharge curves were used to evaluate the degradation of EDLCs [7,8,20,21]; however, the use of the AC impedance method to evaluate the same has not yet been reported. Therefore, we used the AC impedance method to evaluate the degradation of EDLCs herein.

Additionally, an aqueous EDLC was used for the measurements in this experiment. Compared to non-aqueous EDLCs, aqueous EDLCs are characterized by higher safety and power densities, and various electrolytes have been investigated in the past [22,23,24]. However, only few reports on the degradation tests of aqueous EDLCs exist. Therefore, the purpose of this study was to clarify the degradation behavior of aqueous EDLCs for their practical use.

## 2. Materials and Methods

### 2.1. Synthesis of Activated Carbon Electrodes

Activated carbon electrodes were prepared for the degradation tests. Steam-activated carbon (surface area is 2004 $m^{2}/g$ and total pore volume is 1.24 ${\mathrm{cm}}^{3}/g$) [25] (Hosen Co., Osaka, Japan), Ketjen black (surface area is 1445 $m^{2}/g$ and total pore volume is 2.07 ${\mathrm{cm}}^{3}/g$) (EC600JD Ketjen black international Co., Tokyo, Japan), and PTFE (21539-100 Polysciences, Inc., PA, Tokyo, Japan) were mixed in a mass ratio of 8:1:1 to produce the activated carbon electrodes. The mass of each activated carbon electrode was 17.5 mg. The samples were placed in templates, along with nickel meshes as the collecting electrodes, and hot-pressed for 15 minutes at 130 °C and 10 MPa. The final activated carbon electrode was a 10-mm-diameter disc crimped onto a 12-mm square nickel mesh, as shown in Figure 1a. 

We used 8 mol/L KOH (169-20365 FUJIFILM Wako Pure Chemical Co., Osaka, Japan) diluted to 0.5 mol/L as the electrolyte. The activated carbon electrode and stainless steel were used as the cathode and counter electrode, respectively, for the measurements. As the degradation of EDLCs mostly occurred at the cathode [26], this study also aimed to observe factors influencing the degradation. Figure 1b shows the cell used in the measurements. Taking into account the practical usage, airtight and bipolar cells were also used in this study. The activated carbon electrode and counter electrode were set at the right and left sides in the measurement cell, respectively, and the electrolyte was fed into the measurement cell.

### 2.2. Measurement Method

In this study, the frequency dependence of *Z* and *C* of the EDLCs at 25 °C and 80 °C was confirmed and compared using the AC impedance method. A function generator (Model 3990, Keithley, Instruments, Inc., Solon, OH, United States) was used to apply a voltage to the sample. The voltage ($\dot{V}$) was amplified by a voltage amplifier (T-01LGA, Turtle Industry Co., Ltd., Ibaragi, Japan), the current ($\dot{I}$) was amplified by a current-to-voltage conversion amplifier (CA5350, NF Co., Kanagawa, Japan), and the observations were performed using an oscilloscope (TDS3012B, Tektronix, Inc., OR, Tokyo, Japan). From $\dot{V}$ and $\dot{I}$, complex impedance ($\dot{Z}$) and complex capacitance ($\dot{C}$) were derived using the following equations:(1)$$\dot{Z}\;=\;\frac{\dot{V}}{\dot{I}}$$
(2)$$\dot{C}\;=\;\frac{1}{j\omega \dot{Z}}$$The values of Z and C were derived from these equations.

In the high-temperature degradation test, the specimen was measured in a cryostat (LTB250α AS ONE Co., Osaka, Japan), while keeping the temperature constant. The following measurement conditions were considered: maximum applied voltage of 1 V (rated voltage of KOH is 1 V), frequency from 10 mHz to 100 kHz, and 64 measurement points.

## 3. Results

### 3.1. Degradation Tests at 25 °C and 80 °C

The results of the 25 °C and 80 °C degradation tests are shown in Figure 2 and Figure 3, respectively. As observed from the results of the 25 °C degradation test, *C* and *Z* decreased as the frequency increased. It was also evident that the measurements were largely consistent, except for those recorded on the first day of the experiment. The results of the 80 °C degradation test showed that *C* and *Z* decreased as the frequency increased, similar to those observed during the 25 °C degradation test. However, as the number of days increased, the overall *C* decreased while *Z* increased. Additionally, differences were observed in the spectrum from days 1 to 9 and after day 9.

Because voltage dependence of the EDLC was not observed at 1 kHz, the time dependence of Z and C was compared at 1 kHz, and the results are shown in Figure 4. As observed from the measurement results at 25 °C (Figure 4a), C increased and Z decreased until day 5 (black markers); however, the values were approximately constant after day 5.

Conversely, as shown in the results of the 80 °C degradation test (Figure 4b), *C* decreased and *Z* increased as the number of days increased. Rapid degradation was also observed during days 9–11, wherein *C* decreased by approximately 20 μF and *Z* increased by approximately 6 Ω. There are two possible causes for this rapid deterioration: (i)Degradation due to a sudden change in the temperature of the sample;(ii)Degradation due to residual voltage after measurement.

The initial behavior of the EDLC requires clarification to accurately assess the three degradation effects. We considered the process of the electrolyte penetrating the electrode to be the initial behavior observed in this experiment. Additionally, voltage is one of the factors that accelerates the permeation of the electrolyte. Applying a voltage to the EDLC attracts the ions in the electrolyte into the pores and accelerates permeation. Therefore, the results obtained on day 1 were different than those on the other days because of the voltage applied before measurement. Moreover, the voltage applied on day 1 accelerated the permeation of the electrolyte and the measurements obtained after day 2 were constant.

The day when the increase in *C* and decrease in *Z* in the 25 °C degradation test disappeared was used as the date for comparison, i.e., day 5 (black markers), and the reference values were compared to those measured on day 14. The percentage changes obtained from the comparison are shown in Table 1 and Table 2. The final values of *C* and *Z*, as compared to the reference day (day 5), in the degradation test results at 25 °C were 99% (*C* (day 14)/*C* (day 5)) and 98% (*Z* (day 14)/*Z* (day 5)), respectively, and nearly no difference in the rate of change was observed. Meanwhile, *C* decreased to 21.3% while *Z* increased to 847% in the 80 °C environmental degradation test. 

### 3.2. Examination of Degradation by Temperature Change

The first possible cause of the rapid degradation could be a change in the temperature of the sample. The temperature of the cryostat was controlled by water, which evaporates faster at 25 °C than when it is maintained at a constant temperature of 80 °C. The water lost by evaporation was replenished each time, and the water temperature in the cryostat changed simultaneously. In this experiment, the water was replenished once a day, and the resulting temperature change might have influenced the rapid deterioration. Therefore, the deterioration was observed when the water was changed every day. 

The measurement method used for the variable temperature degradation test is shown in Figure 5. The first measurement was performed one day after the sample was placed at 80 °C. After the first measurement, the temperature was reduced to 5 °C, and the second measurement was made one day later. This process was carried out for two weeks, and the deterioration was observed. The measurement results at 80 °C and 5 °C are shown in Figure 6 and Figure 7, respectively. The results were also compared at 1 kHz, and are shown in Figure 8. The amount of change was minor, as compared to that observed in the degradation test at a constant temperature of 80 °C. Therefore, it is unlikely that the temperature change contributed to the rapid degradation.

### 3.3. Investigation of Degradation by Residual Voltage after Measurement

The second possible cause could be the residual voltage after measurement of the current and voltage. In this experiment, a voltage of 1 V, commensurate with the voltage amplitude of the KOH electrolyte solution, was applied to the sample with an applied voltage amplitude of 2 V. We believe that maintaining this voltage could accelerate the degradation of the EDLC. Therefore, experiments were conducted once a day, with and without discharge, after the measurement to compare the results. The measurement results, with and without discharge, are shown in Figure 9 and Figure 10, respectively. Upon comparing the frequency dependence, approximately no change in the spectra with discharge was noted; however, a large variation in the experimental results without discharge was observed. The time dependence of the C and Z spectra at 1 kHz is shown in Figure 11. In the experiment with discharge, the influence of electrolyte penetration was confirmed until day 4, and the measurement results after day 4 were relatively stable. Meanwhile, in the experiment without discharge, C decreased while Z increased. Rapid degradation between days 7–9 was also observed. This behavior accurately corresponds to the rapid degradation that occurred in the 80 °C degradation test.

The measurement on day 4 (black markers) was set as the reference and compared to the rate of change from the measurement on day 14. The results are shown in Table 3 and Table 4. In the case of discharge, the *Z* and *C* values were virtually unchanged, with changes of 105% (*Z* (day 14)/*Z* (day 4)) and 99% (*C* (day 14)/*C* (day 4)), respectively; meanwhile, in the case without discharge, *Z* and *C* varied significantly, and the rates of change were 165% and 61%, respectively. 

Voltage is a strong factor contributing to EDLC degradation as reported previously [16]. A 0.1 V increase in the applied voltage was reported to halve the cycle life, which was closely related to the degradation of the EDLC [9]. This means that even if the voltage is not continuously applied, leaving the voltage is likely to accelerate the deterioration. Additionally, the experimental conditions and results of each degradation test are shown in Table 5 and Table 6. From these results, rapid degradation was observed only when the residual voltages were not removed.

Additionally, memory effect might be involved in this degradation. It has been reported that memory effect is closely related to γ-NiOOH [27]. When Ni electrodes are used, the generation of NiOOH is observed by repeated charging and discharging. In this experiment, as KOH was used as the electrolyte and Ni was used as the collecting electrode, it is highly likely that NiOOH was generated by repeated measurements, and this phenomenon was involved during the rapid degradation.

The deviation, *x*, of the data was obtained from the following equation:(3)$$x=\sqrt{\frac{\sum {\left(A_{i}-\bar{A}\right)}^{2}}{n},}$$
where $A_{i}$ is the *i*-th data point, $\bar{A}$ is the average of the data, and n is the number of data points. The deviations in the results with and without discharge are shown in Table 7. Compared to the case with discharge, the deviation in the case without discharge was observed to be approximately 9 times greater for *C* and 8.5 times greater for *Z*.

## 4. Conclusions

In this experiment, degradation tests for activated carbon EDLCs were conducted at 25 °C and 80 °C. In the 25 °C degradation test, *C* tended to increase while *Z* tended to decrease until day 5, following which the measured values remained nearly constant. This initial behavior represents the process of electrolyte penetration into the electrode. It was confirmed that it took 3–5 days for the electrolyte to completely penetrate the electrode. It is important to understand this initial behavior to clarify the initial values and make accurate measurements.

The results of the 80 °C degradation test confirmed that *Z* increased by a factor of 8 while *C* decreased by a factor of 1/5, as compared to the 25 °C degradation test. It was also confirmed that most of these changes occurred rapidly. There were two possible causes for this change:(i)Degradation due to a sudden change in the temperature of the sample;(ii)Degradation due to residual voltage after measurement.

Under the experimental conditions in this study, we proved that (i) had a limited effect on the rapid degradation; however, (ⅱ) showed a higher possibility of causing rapid degradation and data dispersion. Therefore, it is important to avoid leaving any voltage at the time of measurement. Observing the surfaces of active carbon electrodes and gas analyses are essential future works to further investigate the degradation of EDLCs.

## Figures and Tables

**Figure 1 materials-14-01520-f001:**
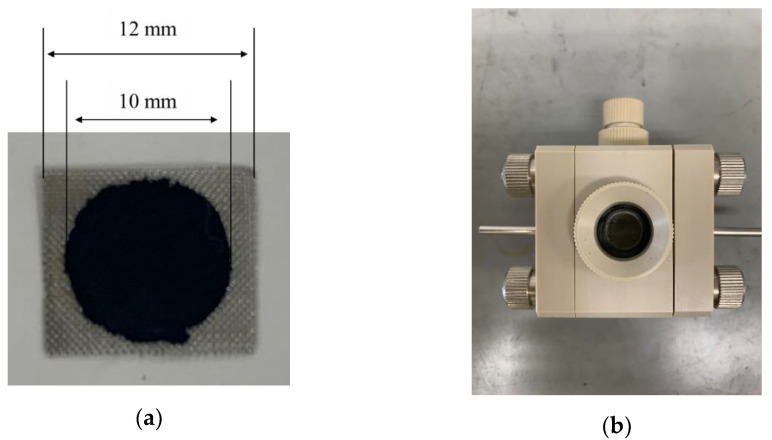
(**a**) Activated carbon electrode and (**b**) measurement cell.

**Figure 2 materials-14-01520-f002:**
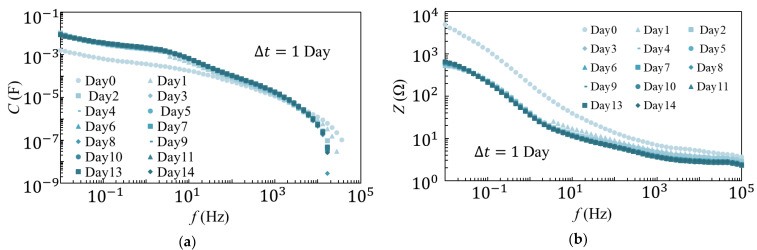
Measurement results of (**a**) *C* and (**b**) *Z* in the 25 °C degradation tests.

**Figure 3 materials-14-01520-f003:**
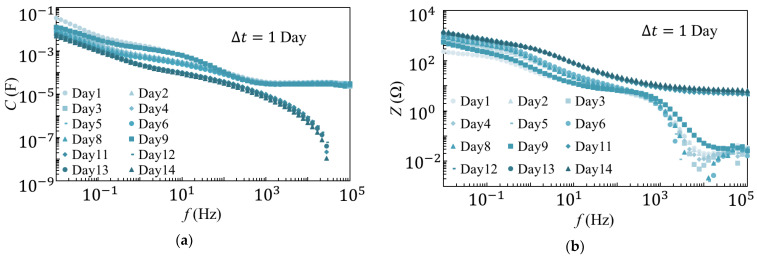
Results of (**a**) *C* and (**b**) *Z* measurements in the 80 °C degradation tests.

**Figure 4 materials-14-01520-f004:**
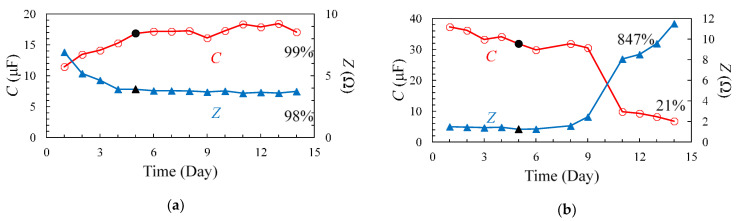
*C* and *Z* characteristics during (**a**) 25 °C and (**b**) 80 °C degradation tests at 1 kHz.

**Figure 5 materials-14-01520-f005:**
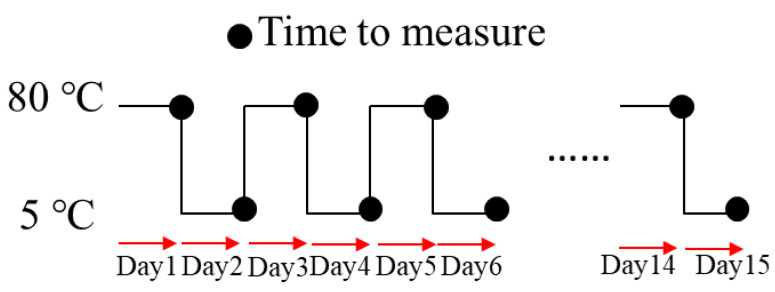
Measurement method for variable temperature degradation tests.

**Figure 6 materials-14-01520-f006:**
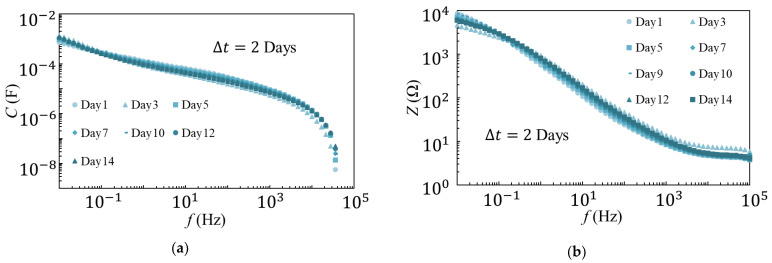
*C* (**a**) and *Z* (**b**) measured at 80 °C in variable temperature (80–5 °C) degradation tests.

**Figure 7 materials-14-01520-f007:**
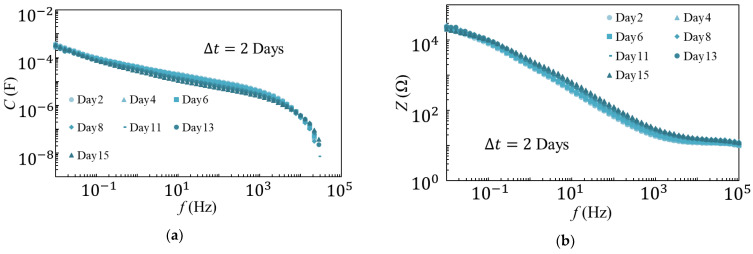
*C* (**a**) and *Z* (**b**) measured at 5 °C in variable temperature (80–5 °C) degradation tests.

**Figure 8 materials-14-01520-f008:**
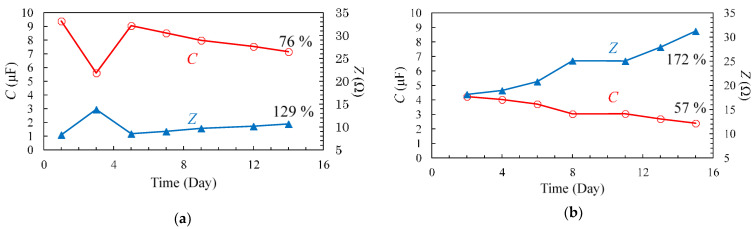
*C* (**a**) and *Z* (**b**) measured at 80 °C in variable temperature (80–5 °C) degradation tests at 1 kHz.

**Figure 9 materials-14-01520-f009:**
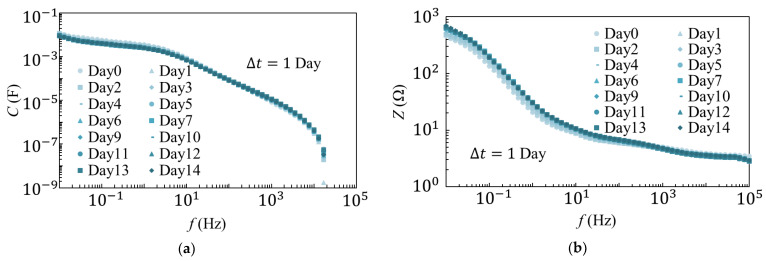
Measurement results of (**a**) *C* and (**b**) *Z* in the 25 °C degradation tests with discharge.

**Figure 10 materials-14-01520-f010:**
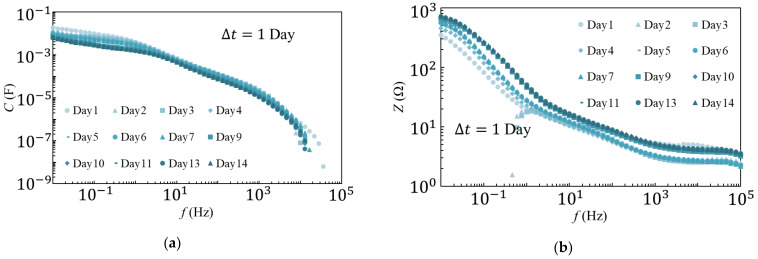
Measurement results of (**a**) *C* and (**b**) *Z* in the 25 °C degradation tests without discharge.

**Figure 11 materials-14-01520-f011:**
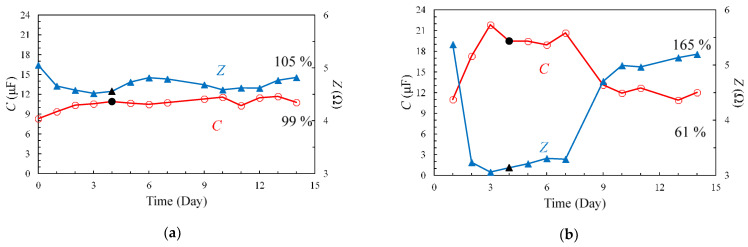
*C* and *Z* characteristics at 1 kHz with (**a**) and without (**b**) discharge.

**Table 1 materials-14-01520-t001:** Percentage change in *C* in the 25 °C and 80 °C degradation tests.

Degradation Test	*C* (Day 5) (μF)	*C* (Day 14) (μF)	Rate of Change in *C* (%)
25 °C	16.8	17.1	99
80 °C	31.8	6.77	21.3

**Table 2 materials-14-01520-t002:** Percentage change in *Z* in the 25 °C and 80 °C degradation tests.

Degradation Test	*Z* (Day 5) (Ω)	*Z* (Day 14) (Ω)	Rate of Change in *Z* (%)
25 °C	3.90	3.74	98
80 °C	1.24	11.5	847

**Table 3 materials-14-01520-t003:** Percentage change in *C* in the degradation tests with and without discharge.

Degradation Test	*C* (Day 4) (μF)	*C* (Day 14) (μF)	Rate of Change in *C* (%)
Discharge	10.9	10.8	99
No discharge	19.5	12.0	61

**Table 4 materials-14-01520-t004:** Percentage change in *Z* in the degradation tests with and without discharge.

Degradation Test	*Z* (Day 4) (Ω)	*Z* (Day 14) (Ω)	Rate of Change in *Z* (%)
Discharge	4.56	4.82	105
No discharge	3.18	5.19	165

**Table 5 materials-14-01520-t005:** Experimental conditions used for degradation tests.

Test Type	Experimental Conditions	
	Temperature (°C)	Remove Residual Voltage
25 °C degradation test	25	Yes
80 °C degradation test	80	-
Variable temperature degradation test	80 ↔ 5	Yes
Residual voltage degradation test	25	-

**Table 6 materials-14-01520-t006:** Results of degradation tests.

Test Type	Results	
	Rapid Degradation	Slow Degradation
25 °C degradation test	-	-
80 °C degradation test	Yes	Yes
Variable temperature degradation test	-	Yes
Residual voltage degradation test	Yes	-

**Table 7 materials-14-01520-t007:** Deviations in measurement results with and without discharge.

Deviation	Discharge	Without Discharge
*C*	$4.25\times 10^{-7}$	$37.3\times 10^{-7}$
*Z*	$1\times 10^{-1}$	$8.48\times 10^{-1}$

## Data Availability

Data sharing not applicable to this article as no datasets were generated.

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
