# Peer review of "High-Temperature Degradation Tests on Electric Double-Layer Capacitors: The Effect of Residual Voltage on Degradation"

_materials, 2021, doi:10.3390/ma14061520_

Round 1

Reviewer 1 Report

The questions from reviewers have been addressed. With the reduction and  the modifications, the quality of the manuscript is much improved. I agree to publish this manuscript.

Reviewer 2 Report

The manuscript entitled: High-temperature degradation tests on electric double-layer capacitors: The effect of residual voltage on degradation deals with the effect of residual voltage on the degradation of double-layer capacitors.

Major Concern:

Please include the setup of the cell, how the supercapacitor was constructed? The cyclability tests are also missing. These data should be included for the reproduction of data by the readers. 

Minor Concern:

  • Typos should be rectified.
  • The English language needs attention preferably by a native English speaker. For instance. Line 176 - ....This behavior corresponded to the.... should be written as  .... this behavior corresponds to the...

Reviewer 3 Report

Following are my comments based on which I suggest a serious and major revision of the manuscript:

Line 14-15 Ketjen black and PTFE are not used as electrode, PTFE is used as binder, and Ketjen black is conductivity additive, please clarify.

Authors should show the gas adsorption properties of carbon material used in this work, what is surface area and pore size. 

Line 71-72 Units of concentration are mol L-1, correct it everywhere.

Line 75 spelling correction > Threat

Line 75-76 Authors must describe the cell in more detail. What kind of experimental setup was used for electrochemical testing. It is important for understanding.

Line 92-93 Authors must explain why they used large voltage of 2 V, which the potential window of KOH based supercapacitor limited to only 0.7-0.8 V.

Line 100-101 a voltage window of 200 mV to 5 V for a KOH based supercapacitor cell? This is not practically possible. Even if it is a degradation test, still applying so high voltage means, all the components are being placed under severe test which may be only correct for a true industrial cell, but I would still doubt such an unrealistic high voltage.   

My main concern is: How authors can apply so high voltage for a supercapacitor which contain KOH electrolyte in a carbon/carbon cell. This point needs to be clarified.

Authors much show galvanostatic charge/discharge curves of this capacitor up to mentioned voltage values in order to see if it is really an electric double layer capacitor (EDLC) as claimed by them.

Figure 1. presentation of this figure 1 is below standard. Authors could make an effort to take a good picture of electrode and detail the caption a little bit more. Such kind of picture can not be part of an international journal publication.

Line 30-34 It is not clear what voltage value authors are talking about.

Round 2

Reviewer 2 Report

The authors have satisfactorily addressed the comments and the manuscript may now be accepted for publication in the present form.

Reviewer 3 Report

Accept

This manuscript is a resubmission of an earlier submission. The following is a list of the peer review reports and author responses from that submission.

Round 1

Reviewer 1 Report

In the proposed manuscript, the authors studied the degradation of electric double layer capacitors. The electrode material presented in this research was active carbon. The authors proposed three possible causes for the degradation, and studied the effects of the three of them on capacitor degradation. Among the three causes, residual voltage was found to be the major cause for capacitance decrease and impedance increase.

There are several aspects need to be improved.

Other than presenting data, the fundamental understanding behind characterization is important as well. In this manuscript, the chemical/physical changes in active carbon were not indicated. Even the degradation behavior is the focus of the study, the explanation should be proposed or referred to previous related research.

Line 84, “80 ËšC and 5 ËšC”, is it “80 ËšC and 25 ËšC”?

Line 92, it should be “it has been studied by He, etc.”

The caption of figure 2 was not completed.

Line 104, it said the data recorded on the first day was not consistent with others. Why?

Line 122, the authors only considered electrolyte penetrating as the initial behavior, is there any other research supporting this? What is the criteria to determine day 5 was the end of initial behavior?

Table 1 should be arranged next to the paragraph on page 4. The actual Z and C data should be presented in the table, not only the percentage.

All the degradation test should be conducted with the same length of time. The vibration test should take long time.

The study about temperature change was confusing. The authors mentioned changing temperature between 80 and 5 ËšC, but figure 7, 8 and 9 all presented data at constant temperature, either 80 or 5 ËšC. The electrode should experience different temperatures multiple times to show the stability with temperature change.

The temperature for the residual voltage study was not clear. The cause for degradation in this part was attributed to the memory effect of nickel only. Is there any data indicating the active carbon did no effort on the degradation?

Line 234, forgot to delete the conclusions section from the template.

Overall, the research was reasonably designed. However, there are some careless mistakes, and the language can be improved. The science behind the data should be included in the manuscript to improve the quality. I would say reconsider this manuscript after Major Revisions.

Reviewer 2 Report

The manuscript entitled: 'Title High-temperature degradation tests of electric double layer capacitor -the effect of residual voltage on degradation' deals with high-temperature degradation tests of electric double-layer capacitors.

  • Scale bar should be introduced in Figure. 1
  • The manuscript involves a lot of experimental data as a function of 14 days. However, it involves very few scientific explanations, where and when required.  
  • It is expected that the electrodes operating at higher temperatures will degrade faster, and no tests are required for the same. At the same time, both current and voltage will act as prominent factors for the degradation. Hence the motivation of the article is missing on a larger picture.
  • Why did the temperature was chosen as 80 C?
  • Typos in the manuscript should be carefully rectified.
  • The English language needs attention.

The manuscript did not advance the science any forward and it may not be recommended for publication in the present form.

Reviewer 3 Report

This article deals with supercapacitors performance at high temperature. However, authors do not provide, any setup of the cell, how supercapacitor was constructed? Cyclic voltammograms and galvanostatic charge/discharge curves are RT and high temperature must be provided. The cyclability tests are also missing. This article requires major improvements before consideration to publish.